# THERMODYNAMIC NATURAL GRADIENT DESCENT

## ABSTRACT

Second-order training methods have better convergence properties than gradient descent but are rarely used in practice for large-scale training due to their computational overhead. This can be viewed as a hardware limitation (imposed by digital computers). Here we show that natural gradient descent (NGD), a second-order method, can have a similar computational complexity per iteration to a first-order method, when employing appropriate hardware. We present a new hybrid digital-analog algorithm for training neural networks that is equivalent to NGD in a certain parameter regime but avoids prohibitively costly linear system solves. Our algorithm exploits the thermodynamic properties of an analog system at equilibrium, and hence requires an analog thermodynamic computer. The training occurs in a hybrid digital-analog loop, where the gradient and Fisher information matrix (or any other positive semi-definite curvature matrix) are calculated at given time intervals while the analog dynamics take place. We numerically demonstrate the superiority of this approach over state-of-the-art digital first- and second-order training methods on classification tasks and language model fine-tuning tasks.

## 1 INTRODUCTION

With the rise of more sophisticated AI models, the cost of training them is exploding, as world-leading models now cost hundreds of millions of dollars to train. This issue is compounded by the ending of both Moore's Law and Dennard's Law for digital hardware (Khan et al., 2018), which impacts both the runtime and energy efficiency of such hardware. This highlights a need and an opportunity for specialized, unconventional hardware targeted at improving the efficiency of training AI models.

Moreover, conventional digital hardware can be viewed as limiting the range of training algorithms that a user may consider. Researchers are missing an opportunity to co-design novel optimizers to exploit novel hardware developments. Instead, relatively simplistic optimizers, such as stochastic gradient descent (SGD), Adam (Kingma & Ba, 2015), and their variants (Loshchilov & Hutter, 2017), are among the most popular methods for training deep neural networks (DNNs) and other large AI models. More sophisticated optimizers are rarely used due to the associated computational overhead on digital hardware.

A clear example of this is second-order methods, which capture curvature information of the loss landscape. These methods, while theoretically more powerful in terms of convergence properties, remain computationally expensive and harder to use, blocking their adoption. For example, natural gradient descent (NGD) (Amari, 1998; Martens, 2020) involves calculating estimates of second-order quantities such as the Fisher information matrix and performing a costly linear system solve at every epoch. Some approximations to NGD, such as the Kronecker-factored approximate curvature (K-FAC) (Martens & Grosse, 2015), have shown promise, and K-FAC has shown superior performance to Adam (Lin et al., 2023; Eschenhagen et al., 2023). However, applying such methods to arbitrary neural network architectures remains difficult (Pauloski et al., 2020).

In this article, we present thermodynamic natural gradient descent (TNGD), a new method to perform second-order optimization. This method involves a hybrid digital-analog loop, where a GPU communicates with an analog thermodynamic computer. A nice feature of this paradigm is flexibility: the user provides their model architecture and the analog computer serves only to accelerate the training process. This is in contrast to many proposals to accelerate the inference workload of AI models with analog computing, where the model is hardwired into the hardware, and users are unable to change the model architecture as they seamlessly would by using their preferred software tools (Kim et al., 2017; Ambrogio et al., 2018; Cristiano et al., 2018; Aguirre et al., 2024).

The analog computer in TNGD uses thermodynamic processes as a computational resource. Such thermodynamic devices have previously been proposed (Conte et al., 2019; Hylton, 2020; Ganesh, 2017; Coles et al., 2023; Lipka-Bartosik et al., 2023), have been theorized to exhibit runtime and energy efficiency gains (Aifer et al., 2023; Duffield et al., 2023), and have been successfully prototyped (Melanson et al., 2023; Aifer et al., 2024). Our TNGD algorithm represents an instance of algorithmic co-design, where we propose a novel optimizer to take advantage of a novel hardware paradigm. TNGD exploits a physical Ornstein–Uhlenbeck process to implement the parameter update rule in NGD. It has a runtime per iteration scaling linearly in the number of parameters, and when properly parallelized it can be close to the runtime of first-order optimizers such as Adam and SGD. Hence, it is theoretically possible to achieve the computational efficiency of a first-order training method while still accounting for the curvature of the loss landscape with a second-order method. Moreover, our numerics show the competitiveness of TNGD with first-order methods for classification and extractive question-answering tasks.

## 2 Related work

There is a large body of theoretical research on natural gradient descent (Amari, 1998; Martens, 2020; Bottou et al., 2018) arguing that NGD requires fewer iterations than SGD to converge to the same value of the loss in specific settings. While less is known about the theoretical convergence rate of Adam, there exists a large body of empirical evidence that NGD can converge in fewer iterations than Adam (Martens et al., 2010; Martens & Grosse, 2015; Martens et al., 2018; Eschenhagen et al., 2023; Ren & Goldfarb, 2019; Gargiani et al., 2020).

However, a single iteration of NGD is generally more computationally expensive than that of SGD or Adam, which have a per-iteration cost scaling linearly in the number of parameters $N$. NGD typically has a superlinear (assuming the condition number scales as $\kappa = N^\alpha, \alpha > 0$ for NGD-CG) complexity in the number of parameters (although this may be reduced to linear scaling at the expense of higher-order scaling with batch size and output dimension, see Section 3). K-FAC (Martens & Grosse, 2015) aims to reduce this complexity and invokes a block-wise approximation of the curvature matrix, which may not always hold. While first introduced for multi-layer perceptrons, K-FAC has been applied to more complex architectures, such as recurrent neural networks (Martens et al., 2018) and transformers (Eschenhagen et al., 2023), where additional approximations have to be made and where the associated computational overhead can vary.

There has been significant effort and progress towards reducing the time- and space- complexity of operations used in the inference workload of AI models, e.g., a variety of "linear attention" blocks have been proposed (Shen et al., 2021; Katharopoulos et al., 2020; Wang et al., 2020). However, there has been less focus on reducing the complexity of training methods. While various approaches are taken to accelerating training using novel hardware, these efforts typically aim at reducing the constant coefficients appearing in the time cost of computation. Especially relevant to our work, analog computing devices have been proposed to achieve reduced time and energy costs of training relative to available digital technology (Kim et al., 2017; Ambrogio et al., 2018; Cristiano et al., 2018; Aguirre et al., 2024). These devices are generally limited to training a neural network that has a specific architecture (corresponding to the structure of the analog device). To our knowledge, there has not yet been a proposal that leverages analog hardware to reduce the complexity of training algorithms such as NGD.

Given the existing results implying that fewer iterations are needed for NGD relative to other commonly used optimizers, we focus on reducing the per-iteration computational cost of NGD using a hybrid analog-digital algorithm to perform each parameter update. Our algorithm therefore demonstrates that complexity can be improved in training (not only in inference), and moreover that the per-iteration complexity of NGD can be made similar to that of a first-order training method.

## 3 Natural gradient descent

Let us consider a supervised learning setting, where the goal is to minimize an objective function defined as:

$$\ell(\theta) = \frac{1}{|\mathcal{D}|} \sum_{(x,y) \in \mathcal{D}} L(y, f_\theta(x)), \tag{1}$$

where $L(y, f_\theta(x)) \in \mathbb{R}$ is a loss function, $f_\theta(x)$ is the forward function that is parametrized by $\theta \in \mathbb{R}^N$. These functions depend on input data and labels $(x, y) \in \mathcal{D}$, with $\mathcal{D}$ a given training dataset. Viewed through the lens of statistics, minimizing the objective function is analogous to minimizing the Kullback-Leibler (KL) divergence from the target joint distribution $q(x, y)$ to the learned distribution $p(x, y|\theta)$ (Martens, 2020). A straightforward way to optimize $\ell(\theta)$ is to follow the direction of steepest descent, defined by the negative gradient $-\nabla \ell$, defined as:

$$\frac{-\nabla \ell}{||\nabla \ell||} = \lim_{\epsilon \to 0} \frac{1}{\epsilon} \underset{d:||d|| \leq \epsilon}{\arg\min} \ell(\theta + d), \tag{2}$$

with $|| \cdot ||$ the Euclidean norm. The natural gradient, on the other hand can be defined as the direction of steepest descent with respect to the KL divergence defined as:

$$\text{KL}(p(x, y|\theta + d)||p(x, y|\theta)) = \iint p(x, y|\theta + d) \, \log \left( \frac{p(x, y|\theta + d)}{p(x, y|\theta)} \right) \, \mathrm{d}x \mathrm{d}y \tag{3}$$

(Amari & Nagaoka, 2000). One may then Taylor-expand this divergence as

$$\text{KL}(p(x, y|\theta + d)||p(x, y|\theta)) = \frac{1}{2} d^\top F d + O(d^3), \tag{4}$$

where $F$ is the *Fisher information matrix* (Martens, 2020) (or the *Fisher*), defined as:

$$F = \mathbb{E}_{p(x,y|\theta)}[\nabla \log p(x, y|\theta) \nabla \log p(x, y|\theta)^\top]. \tag{5}$$

The natural gradient is then simply defined as

$$\tilde{g} = F^{-1} \nabla \ell(\theta). \tag{6}$$

For the NGD optimizer, the update rule is then given by:

$$\theta_{k+1} = \theta_k - \eta F^{-1} \nabla \ell, \tag{7}$$

with $\eta$ a learning rate. In practice, computing the Fisher information is not always feasible because one must have access to the density $p(x, y|\theta)$. A quantity that is always possible (and relatively cheap) to compute thanks to auto-differentiation is the empirical Fisher information matrix, defined as:

$$\bar{F} = JJ^\top = \frac{1}{b} \sum_{(x,y) \in \mathcal{S}} \nabla \log p(y|x, \theta) \nabla \log p(y|x, \theta)^\top, \tag{8}$$

where $\log p(y \mid x, \theta) = -L(y, f_\theta(x))$, $|\mathcal{S}| = b$ is the batch size and $\mathcal{S} \subset \mathcal{D}$. The Jacobian matrix $J$ is defined as

$$J = \frac{1}{\sqrt{b}}[\nabla \log p(y_1|x_1, \theta), \nabla \log p(y_2|x_2, \theta), \dots, \nabla \log p(y_b|x_b, \theta)].$$

Note that the squared gradient appearing in the second moment estimate of the Adam optimizer (Kingma & Ba, 2015) is the diagonal of the empirical Fisher matrix. Another approximation to the Fisher matrix is the generalized Gauss-Newton (GGN) matrix, defined as:

$$G = J_f H_L J_f^\top = \frac{1}{b} \sum_{(x,y) \in \mathcal{S}} J_f^{(x,y)} H_L^{(x,y)} J_f^{(x,y)\top}, \tag{9}$$

where $J_f^{(x,y)}$ is the Jacobian of $f_\theta(x)$ with respect to $\theta$ and $H_L^{(x,y)}$ is the Hessian of $L(y, z)$ with respect to $z$ evaluated at $z = f_\theta(x)$. $J_f$ is a $bd_z \times N$ matrix, and $H_L$ is a $bd_z \times bd_z$ matrix, where $d_z$ is the output dimension of $z = f_\theta(x)$ and $N$ is the number of parameters ($N$ also depends on $d_z$, where for deep networks it is a weak dependence).

For loss functions of the exponential family (with natural parameter $z$), the GGN matches the true Fisher matrix (Martens, 2020). In addition, we have observed better convergence with the GGN than with the empirical Fisher (as in other works such as Refs. Martens et al. (2010); Kunstner et al. (2019), where better convergence than with the Hessian is also observed). Therefore, we will consider the GGN in what follows. Note that the methods we introduce in this work apply to any second-order optimization algorithm with a positive semi-definite curvature matrix (by curvature matrix, we mean any matrix capturing information about the loss landscape). In particular, it applies most efficiently to matrices constructed as outer products of rectangular matrices (such as the empirical Fisher and the GGN) as explained below.

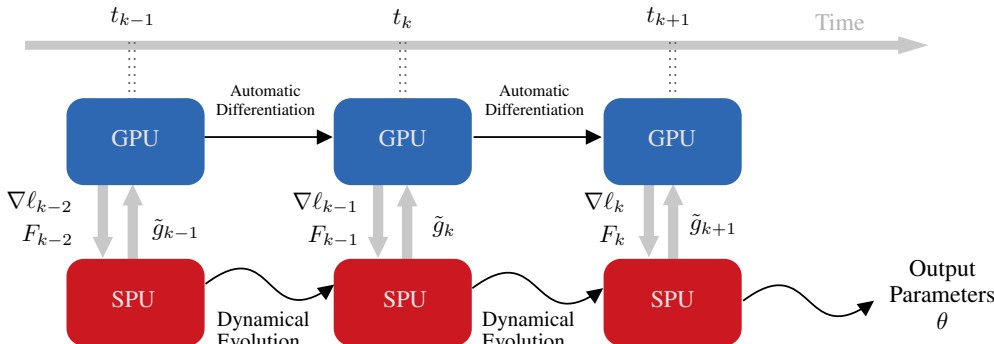

Figure 1: **Overview of Thermodynamic Natural Gradient Descent (TNGD).** A GPU that stores the model architecture and provides the gradient $\nabla\ell_k$ and Fisher matrix $F_k$ (through its representation given by the Jacobian $J_f$ and Hessian $H_L$ matrices given by equation 9) at step $k$ is connected to a thermodynamic computer, called the stochastic processing unit (SPU). At times $t_k$, the estimate of the natural gradient $\tilde{g}_k$ is sent to the GPU, which updates the parameters of the model and calculates gradients and curvature matrices for some new data batch $(x_k, y_k)$. During digital auto-differentiation, the SPU undergoes dynamical evolution, either continuing to approach its steady-state or remaining in it. After some time, gradient $\nabla\ell_k$ and Fisher matrix $F_k$ are sent to the SPU through a DAC and digital controllers. This modifies the dynamics of the SPU, and after some time interval, a new natural gradient estimate $\tilde{g}_{k+1}$ is sent back to the GPU. Note that the time between two measurements $t_{k+1} - t_k$ need not be greater than the time between two auto-differentiation calls. The hybrid digital-thermodynamic process may be used asynchronously as shown in the diagram (where the time of measurement of $\tilde{g}$ and upload of the gradient and Fisher matrix are not the same).

## 3.1 FAST MATRIX VECTOR PRODUCTS

The linear system appearing in equation 6 can be solved using the conjugate gradient (CG) method (Martens et al., 2010), which will be referred to as NGD-CG in what follows. In fact, when $\ell$ is parametrized by a neural network, the GGN-vector product $Gv$ involved in the conjugate gradient algorithm may be evaluated in runtime $O(bN)$ thanks to fast Jacobian-vector products (Bradbury et al., 2018) (JVPs). This approach also enables one to not explicitly construct the Fisher matrix, thus also avoiding a $O(bd_z N^2)$ runtime cost in computing it and a $O(N^2)$ memory cost in storing it. The efficiency of this approach depends on the number of CG iterations required to obtain good performance. Importantly, convergence in $\sqrt{\kappa}$ steps, with $\kappa$ the condition number of $F$, is not required to obtain competitive performance (Martens & Grosse, 2015; Gargiani et al., 2020). Crucially, due to the sequential nature of the algorithm, the CG iterations cannot be parallelized.

In practice, since reaching convergence is computationally expensive, one generally stops the CG algorithm after a set number of iterations. Because of the way the step size is adapted in CG, we have observed that the solution after $k$ steps $x_k$ is not necessarily closer to the true solution than the initial guess $x_0$, in particular for ill-conditioned problems, which can make NGD-CG difficult to use.

## 3.2 NGD WITH THE WOODBURY IDENTITY

In the machine learning setting, it is often the case that $b \ll N$ (and $d_z \ll N$). This means that the curvature matrix is low-rank and the linear system to solve is underdetermined. To mitigate this issue, the Fisher matrix may be dampened as $F + \lambda\mathbb{I}$. In that case, the Woodbury identity may be used to obtain the inverse Fisher vector-product $F^{-1}v$ appearing in the NGD update. We have:

$$F = UV + \lambda\mathbb{I}, \text{ with } U = J_f, V = H_L J_f^\top \tag{10}$$

$$F^{-1} = \lambda^{-1}\mathbb{I} - \lambda^{-2}U(\mathbb{I} + \lambda^{-1}VU)^{-1}V \quad \text{(Woodbury)} \tag{11}$$

$$F^{-1}v = \lambda^{-1}\mathbb{I} - \lambda^{-2}U(\mathbb{I} + \lambda^{-1}VU)^{-1}Vv \tag{12}$$

This is included in Ren & Goldfarb (2019), and can be competitive with NGD-CG when the batch size $b$ and output dimension $d_z$ are much smaller than the number of trainable parameters $N$. Here

| **Optimizer** | Runtime | Memory | Model calls |
|---|---|---|---|
| SGD/Adam | $O(bN)$ | $O(N)$ | 1 |
| NGD | $O(N^3 + bd_z N^2)$ | $O(N^2)$ | $bd_z$ |
| NGD-CG | $O(cbN)$ | $O(N)$ | $2c$ |
| NGD-Woodbury | $O(bd_z^2 N + b^3 d_z^3)$ | $O(bd_z N + bd_z^2)$ | $bd_z$ |
| Thermodynamic NGD | $O(bd_z N + t)$ | $O(bd_z N + bd_z^2)$ | $bd_z$ |

Table 1: **Runtime and memory complexity of optimizers considered in this paper.** All operations are per iteration. The first line corresponds to first-order optimizers that evaluate the gradient only, and apply diagonal rescalings and $O(N)$ operations to it only. Vanilla NGD (second line) includes the explicit storage and inversion of the GGN matrix as well as its construction, dominating the runtime and memory cost. NGD-CG (third line) can be performed by running $c$ iterations, each dominated by GGN-vector products and has the same memory cost as first-order methods. NGD-Woodbury can be performed by constructing the matrix $VU$, and using the formula given by equation 12. This results in a runtime cost dominated by constructing $VU$ and inverting it, which also requires its storage.

one must construct the $V$ matrix, which has runtime $O(d_z^2 bN)$ (since $H_L$ is block-diagonal), and invert $(\mathbb{I} + \lambda^{-1} VU)$ which is $O(b^3 d_z^3)$. While the batch size typically remains small, the value of $d_z$ can make this inversion intractable. For example, in many language-model tasks, $d_z \sim O(10^4)$ is the vocabulary size.

## 4 THERMODYNAMIC NGD

At a high level, TNGD combines the strength of GPUs (through auto-differentiation) with the strength of thermodynamic devices at solving linear systems. Regarding the latter, Aifer et al. (2023) showed that a thermodynamic device, called a stochastic processing unit (SPU), can solve a linear system $Ax = b$ with reduced computational complexity relative to standard digital hardware. The solution to the linear system is found by letting the SPU evolve under an Ornstein–Uhlenbeck (OU) process given by the following stochastic differential equation (SDE):

$$dx = -(Ax - b)dt + \mathcal{N}\left[0, 2\beta^{-1}\, dt\right], \tag{13}$$

where $A$ is a positive matrix and $\beta$ is a positive scalar (which can be seen as the inverse temperature of the noise). Operationally, one lets the SPU settle to its equilibrium state under the dynamics of equation 13, at which point $x$ is distributed according to the Boltzmann distribution given by:

$$x \sim \mathcal{N}[A^{-1}b, \beta^{-1}A^{-1}]. \tag{14}$$

One can see that the first moment of this distribution is the solution to the linear system $Ax = b$. Exploiting this approach, TNGD involves a subroutine that estimates the solution to the linear system in equation 6. For this particular linear system, the SDE in equation 13 becomes the following:

$$d\tilde{g}_{k,t} = -(F_{k-1}\tilde{g}_{k,t} - \nabla\ell_{k-1})dt + \mathcal{N}\left[0, 2\kappa_0 dt\right] \tag{15}$$

$$= -(J_{f,k-1}^\top H_{L,k-1} J_{f,k-1}\tilde{g}_{k,t} - \nabla\ell_{k-1})dt + \mathcal{N}\left[0, 2\kappa_0 dt\right] \tag{16}$$

with $\tilde{g}_{k,t}$ the value of the natural gradient estimate at time $t$ and $\kappa_0$ the variance of the noise. Comparing equation 13 and equation 15, we see that in the equilibrium state (i.e. for large $t$), the mean of $\tilde{g}_{k,t}$ provides an estimate of the natural gradient, in other words:

$$\tilde{g}_k := \lim_{t \to \infty} \langle \tilde{g}_{k,t} \rangle = F_{k-1}^{-1} \nabla\ell_{k-1}. \tag{17}$$

The overall TNGD algorithm is illustrated in Fig. 1. Using the current parameter estimates $\theta_k$, the GPU computes the matrices $J_f$ and $H_L$, and the vector $\nabla\ell$, which can be accomplished efficiently using auto-differentiation. The matrices $J_f$, $J_f^\top$, and $H_L$, as well as the vector $\nabla\ell$, are uploaded to component values (see Appendix) on the SPU, which is then allowed to equilibrate under the dynamics of equation 15. Next, samples are taken of $\tilde{g}_{t,k}$, and are sent from the SPU to the GPU,

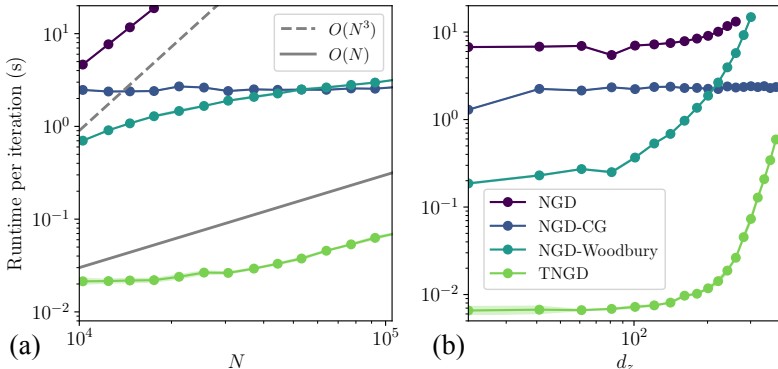

Figure 2: **Runtime per iteration of second-order optimizers considered in this paper.** (a) The runtimes per iteration are compared for NGD, NGD-CG, NGD-Woodbury, and TNGD (estimated) for various $N$. Here the convolutional network we applied to MNIST is used and the dimension of the hidden layer is varied to vary $N$ for fixed $d_z = 20$. (b) The same comparison is shown for various values of $d_z$. The same network is used and $d_z$ is varied (this also has the effect of varying the $N$). Error bars are displayed as shaded area but are smaller than the data markers.

where samples are averaged to yield an estimate of $\tilde{g}_k$. Finally, the parameters are updated using the equation

$$\theta_{k+1} = \theta_k - \eta \tilde{g}_k, \tag{18}$$

and this process may be repeated until sufficient convergence is achieved (other update equations may also be employed, see Section 5).

While equation 17 involves a long time limit, numerical evidence (see Section 5) shows that samples may be taken even before equilibrium has been reached without harming performance significantly. Thus, the analog dynamics time $t$ is an important hyperparameter of TNGD. Furthermore, another hyperparameter arises from the delay time $t_d$, defined as the time between a measurement of $\theta_k$ and the update of the gradient and GGN on the device. As discussed in Section 5, a non-zero delay time is not necessarily detrimental to performance and can in fact improve it.

In addition to the advantage in time- and energy-efficiency, TNGD has another advantage over NGD-CG in terms of stability. For some pathological linear systems, CG fails to converge and instead diverges. However, the thermodynamic algorithm is guaranteed to converge (on average) for any positive definite matrix. To see this, note that the mean of $\tilde{g}_{k,t}$ evolves according to

$$\langle \tilde{g}_{k,t} \rangle = \exp\left(-F_{k-1}t\right)(\tilde{g}_{k,0} - F_{k-1}^{-1}\nabla\ell_{k-1}) + F_{k-1}^{-1}\nabla\ell_{k-1}. \tag{19}$$

There is still variance associated with the estimator of $\langle \tilde{g}_{k,t} \rangle$ (the sample mean), but the sample mean converges to the solution with high probability in all cases. We also note that if we choose $\tilde{g}_{k,0} = \nabla\ell_{k-1}$, we obtain a smooth interpolation between SGD ($t = 0$) and NGD ($t = \infty$).

### 4.1 COMPUTATIONAL COMPLEXITY AND PERFORMANCE

The runtime complexity of TNGD and other second-order optimization (that do not make assumptions on the structure of $G$, hence excluding K-FAC) algorithms is reported in Table 1. As explained, Thermodynamic NGD (TNGD) has a runtime and memory cost dominated by the construction and storage (before sending them off to the analog hardware) of the Jacobian of $f_\theta(x)$ and the Hessian of the loss. The $t$ factor denotes the analog runtime, and may be interpreted similarly to $c$ for NGD-CG as a parameter controlling the approximation. For each optimizer the number of model calls is reported. For all optimizers except NGD-CG these calls can be easily parallelized thanks to vectorizing maps in PyTorch.

In Fig. 2 a comparison of the runtime per iteration of the four second-order optimizers considered is shown. Fig. 2(a) shows the runtime as a function of the number of parameters $N$. The scaling of NGD as $N^3$ can be observed, and the NGD-CG data is close to flat, meaning the model calls parallelize well for the range of parameter count considered. The linear scaling of NGD-Woodbury and TNGD

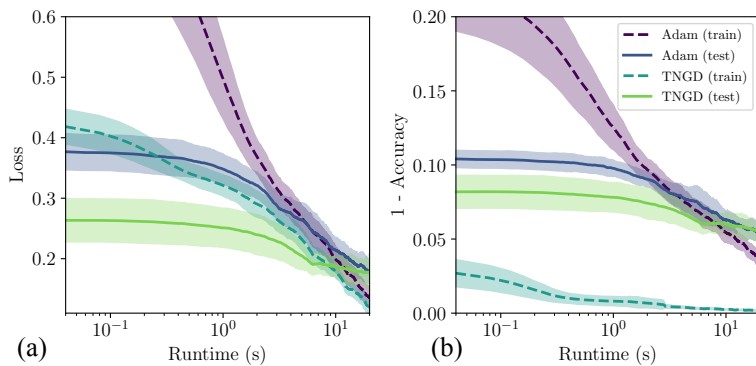

Figure 3: **Performance comparison of Adam and TNGD (estimated) on MNIST classification**. (a) Training (dashed lines) and test loss (solid lines) for Adam (darker colors) and TNGD (lighter colors) are plotted against runtime (measured for Adam, and estimated for TNGD from the timing model described in Section 4.1). Shaded areas are standard deviations over five random seeds. Note that Adam includes adaptive averaging of first and second moment estimates with $(\beta_1, \beta_2) = (0.9, 0.999)$, while TNGD does not. (b) $1 - \text{Accuracy}$ for training and test sets.

is also shown, although with a different overall behaviour due to parallelization and a much shorter runtime per iteration for TNGD. This shows that for the given range of $N$ at $d_z = 20$, we can expect a $100\times$ speedup over second-order optimizers. Fig. 2(b) shows the dependence of runtime on the output dimension $d_z$ for the second-order optimizers. These results indicate that TNGD is most competitive for intermediate values of $d_z$. Finally we note that with better hardware, the scaling with both $N$ and $d_z$ would be better, as the operations to construct the Hessian and Jacobian can be more efficiently parallelized for larger values.

## 5 EXPERIMENTS

### 5.1 MNIST CLASSIFICATION

We first consider the task of MNIST classification (LeCun, 1998). For our experiments, we use a simple convolutional neural network consisting of a convolutional layer followed by two feedforward layers, and we digitally simulate the TNGD algorithm (see App. D). The goal of these experiments is twofold: (1) to compare the estimated performance per runtime of TNGD against popular first-order optimizers such as Adam, and (2) to provide some insights on other features of TNGD, such as its performance as a function of the analog runtime $t$ as well as its asynchronous execution as a function of the delay time $t_d$.

In Fig. 3(a), the training and test losses as a function of runtime for both Adam (measured) and TNGD (estimated) are presented. To estimate the TNGD runtime, we took into account results for its runtime per iteration as presented in the previous section, finding an overall $2\times$ runtime per iteration with respect to Adam for this problem on an A100 GPU. One can see from the figure that even while taking into account the difference in runtime per iteration, TNGD still outperforms Adam, especially at the initial stages of the optimization. Interestingly, it also generalizes better for the considered experimental setup. In Fig.3(b), the training and test accuracies are shown. We again see TNGD largely outperforming Adam, reaching the same training accuracy orders of magnitude faster, while also displaying a better test accuracy. These results are reminiscent of prior work on NGD (Martens et al., 2010), however here the batch size is smaller than in other works, indicating that even a noisy GGN matrix improves the optimization.

As mentioned previously, the continuous-time nature of TNGD allows one to interpolate smoothly between first- ($t = 0$) and second- ($t = \infty$) order optimization, with a given optimizer choice (whether the optimizer update rule is that of SGD or that of Adam as described in Alg. 1). In Fig. 4(a), the training loss vs. iterations is shown for various analog dynamics times. These results clearly demonstrate the effect mentioned above, where increasing the analog runtime improves

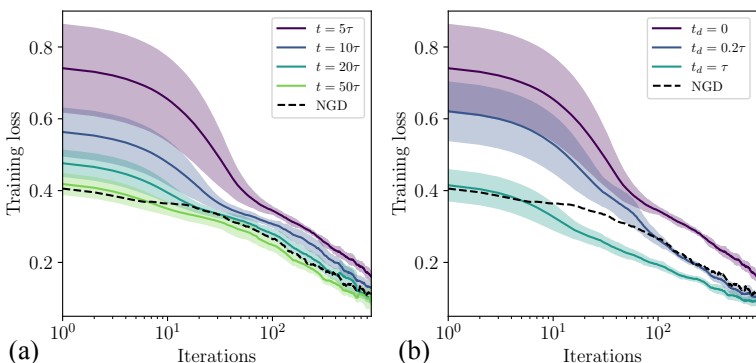

(a)      (b)

Figure 4: **Training loss vs. iterations for varying analog dynamics times**. (a) The training loss is shown for NGD (dashed line) and for TNGD with various analog dynamics times $t$ (solid lines). (b) The training loss is shown for NGD (dashed line) and for TNGD with fixed analog dynamics time $t = 5\tau$ and varying delay times $t_d$ (solid lines). The delay appears to have a momentum effect, which can even lead to TNGD outperforming exact NGD for certain analog dynamics and delay times. Shaded areas are standard deviations over five random seeds.

performances continuously until it approaches that of exact NGD for $t \sim 50\tau$. In Fig. 4(b), the same quantity is shown for a fixed analog dynamics time $t$, and varying delay times $t_d$. This leads to a quadratic approximation of the objective function that is inaccurate (since the GGN and gradients are calculated for parameters different than the value around which the objective function is approximated). However, this results in an improved performance, even for a small delay time. A likely explanation of this result is that the state of the device retains information about the curvature of the previous quadratic approximation, while being subject to the updated quadratic approximation. This effect propagates across iterations which is reminiscent of momentum.

## 5.2 LANGUAGE MODEL FINE-TUNING

In this section we show how thermodynamic NGD may be applied to language modeling tasks, in more practically relevant settings than MNIST classification. We consider the DistilBert model (Sanh et al., 2019) which we fine-tune on the Stanford question-answering dataset (SQuaD) (Rajpurkar et al., 2016), a common dataset to evaluate model comprehension of technical domains through extractive question-answering. As is commonly performed when fine-tuning, we apply a low-rank adaptation (Hu et al., 2021) to the model, which reduces its trainable parameters (details about this procedure are in App. E) to a manageable amount ($75k$ here) for limited compute resources.

Figure 5(a) displays a comparison of the training loss for different optimizers. The bare TNGD (as used in the previous section) shows a worse performance than Adam in this setting. However, a hybrid approach, TNGD-Adam, where the natural gradient estimate is used in conjunction with the Adam update rule gives the best performance (this is explained in App. B). One possible explanation for this result is that there are two pre-conditionings of the gradient for TNGD-Adam: the first comes from the natural gradient, which incorporates curvature information, and the second comes from the Adam update rule, which acts as a signal-noise ratio as explained in Kingma & Ba (2015), which further adjusts the natural gradient values. In Fig. 5(b), we show that the same results as in the previous section apply to TNGD-Adam, where increasing the analog runtime boosts performance. Therefore, the analog runtime in TNGD may be viewed as a resource in this sense, that is computationally very cheap (as time constants can be engineered to be very small).

## 6 LIMITATIONS

The practical impact of our work relies on the future availability of analog thermodynamic computers, such as a scaled up version of the system in Melanson et al. (2023). We provide a circuit diagram of a potential thermodynamic computer in the Appendix. Such computers can employ standard

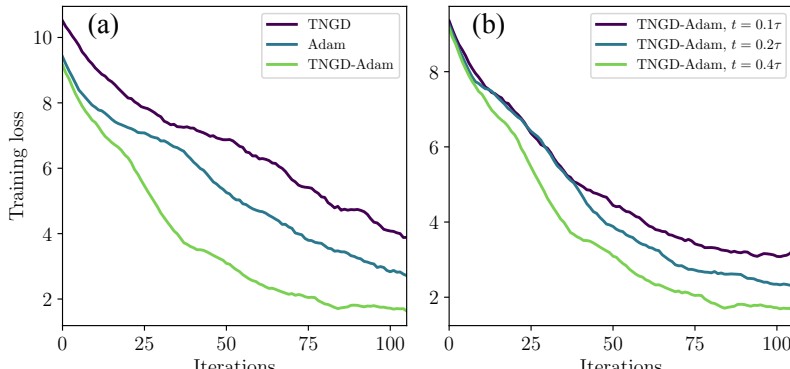

Figure 5: **Training loss vs. iterations for QA fine-tuning.** (a) Comparison of the performance per iteration of TNGD, Adam, and TNGD-Adam, where the latter uses the natural gradient estimate in conjunction with the Adam update rule with $(\beta_1, \beta_2) = (0, 0)$. (b) Performance of the TNGD-Adam optimizer for various analog dynamics times. Similar to Fig. 4, the performance improves as $t$ grows.

electrical components and leverage CMOS-based fabrication infrastructure, and hence are likely straightforward to scale up, although that remains to be demonstrated.

Analog computers, in general, tend to face precision issues, whereby the solution accuracy is limited by the precision of the electrical components. For analog thermodynamic computers, it is possible to mitigate this issue through an averaging technique (Aifer et al., 2024), and the method proposed in Aifer et al. (2024) can be directly applied to the TNGD algorithm to improve solution accuracy. Nevertheless, we suspect that training-based applications will have a significant tolerance to precision-based errors, although a detailed study is needed to confirm that hypothesis. We note that there is a growing body of work on very low-precision inference (Ma et al., 2024) and training (Sun et al., 2020) which indicates that high numerical precision is not crucial for good performance in machine learning. We also remark that thermodynamic computers are predicted to be robust to stochastic noise sources since stochasticity is a key component of such computers (Coles et al., 2023), as is shown in Fig. 6 in the Appendix.

We have numerically tested TNGD for a small subset of potential tasks such as MNIST classification and DistilBert fine-tuning on the SQuaD dataset, for a small number of epochs. Hence, seeing if the advantage we observe for TNGD also holds for other applications is an important direction.

## 7 CONCLUSION

This work introduced Thermodynamic Natural Gradient Descent (TNGD), a hybrid digital-analog algorithm that leverages the thermodynamic properties of an analog system to efficiently perform second-order optimization. TNGD greatly reduces the computational overhead typically associated with second-order methods for arbitrary model architectures. Our numerical results on MNIST classification and language model fine-tuning tasks demonstrate that TNGD outperforms state-of-the-art first-order methods, such as Adam, and provide large speedups over other second-order optimizers. This suggests a promising future for second-order methods when integrated with specialized hardware.

Looking forward, our research stimulates further investigation into TNGD, particularly with enhancements such as averaging techniques and moving averages. Extensions to approximate second-order methods such as K-FAC may also be possible. Moreover, the principles of thermodynamic computing could inspire new algorithms for Bayesian filtering. While the current impact of our work relies on the development and availability of large-scale analog thermodynamic computers, the theoretical and empirical advantages presented here underscore the potential of co-designing algorithms and hardware to overcome the limitations of conventional digital approaches.

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

## A    LEVENBERG-MARQUARDT REGULARIZATION SCHEDULE

Poor-conditioning and singularity of the curvature matrix can greatly decrease the performance of NGD, which is dealt with by adding a term $\lambda \mathbb{I}$ to the curvature matrix. Following  Martens et al. (2010), it is possible to use a simple method to adapt the value of $\lambda$ at each time step, known as a Levenberg-Marquardt schedule. This involves computing the reduction ratio $\rho$, defined as:

$$\rho = \frac{\ell(\theta_{k+1}) - \ell(\theta_k)}{q_{\theta_k}(\theta_{k+1} - \theta_k) - q_{\theta_k}(0)} \tag{20}$$

with $q_{\theta_k}(p)$ the quadratic approximation to $\ell$ around $\theta_k$ defined as:

$$q_{\theta_k}(p) = \ell(\theta_k) + \nabla \ell(\theta_k)^\top p + \frac{1}{2} p^\top G_k p. \tag{21}$$

If $\rho > a$, $\lambda \leftarrow \alpha \lambda$, and if $\rho < 1 - a$, $\lambda \leftarrow \lambda/\alpha$. This can be interpreted as distrusting the quadratic model when $\rho$ is small, hence increasing $\lambda$ for the next iteration. This procedure may be used for TNGD, however this adds a supplementary digital cost of a GGN-vector product $G_k p$ (which has a similar cost to two JVPs). For our experiments we did not find it to significantly boost performance although it may be considered for future work.

## B    TNGD ALGORITHM

In Alg. 1 we provide the steps for the TNGD algorithm. This algorithm may be used in conjunction with various digital optimizers (such as SGD or Adam). The thermodynamic linear solver (TLS) is performed by an analog thermodynamic computer whose physical implementation is described in appendix C. The TLS takes as inputs the Jacobian $J_{f,k}$, the Hessian $H_L$, the gradient $g_k$ and an initial point $x_0$ (that can be reset at each iteration, or not, in which case $t_d > 0$).

---

**Algorithm 1** Thermodynamic Natural Gradient Descent

---

**Require:** $n > 0$
    Initialize $\theta_0$
    $\tilde{g}_0 \leftarrow \nabla \ell(\theta_0)$
    $\texttt{optimizer} \leftarrow \texttt{SGD}(\eta, \beta)$ **or** $\texttt{Adam}(\eta, \beta_1, \beta_2)$
    **while** $k \neq n$ **do**
        $x_k, y_k \leftarrow$ next batch
        $g_k \leftarrow \nabla \ell(\theta_k, x_k, y_k)$
        $\tilde{g}_k \leftarrow \texttt{TLS}(J_f, H_L, b = g_k, x_0 = \tilde{g}_{k-1})$
        $\texttt{optimizer.update}(\theta_k, \tilde{g}_k)$
        $k \leftarrow k + 1$
    **end while**

---

## C    HARDWARE IMPLEMENTATION

The thermodynamic NGD algorithm can be implemented in similar hardware to what is presented in Refs. Aifer et al. (2023); Melanson et al. (2023). However, this requires one to construct the full curvature matrix, which is quadratic in the number of parameters, and then send it to the analog hardware. Therefore, an alternative hardware implementation that is described by the same electronic circuit equations is preferred.

This alternative implementation is comprised of three arrays of resistors of size $(bd_z, N)$, $(bd_z, bd_z)$ and $(N, bd_z)$ for storing $J_f$, $H_L$ and $J_f^\top$, respectively (hence two of these are rectangular resistor arrays). These three arrays of resistors enable one to implement the following differential equation in hardware:

$$dV = -(J_f H_L J_f^\top + \lambda \mathbb{I})V dt - b dt + \mathcal{N}(0, 2\kappa_0 dt) \tag{22}$$

where $\kappa_0$ is the noise variance and $V = (V_1, V_2, \ldots, V_N)$ is a vector of voltages.

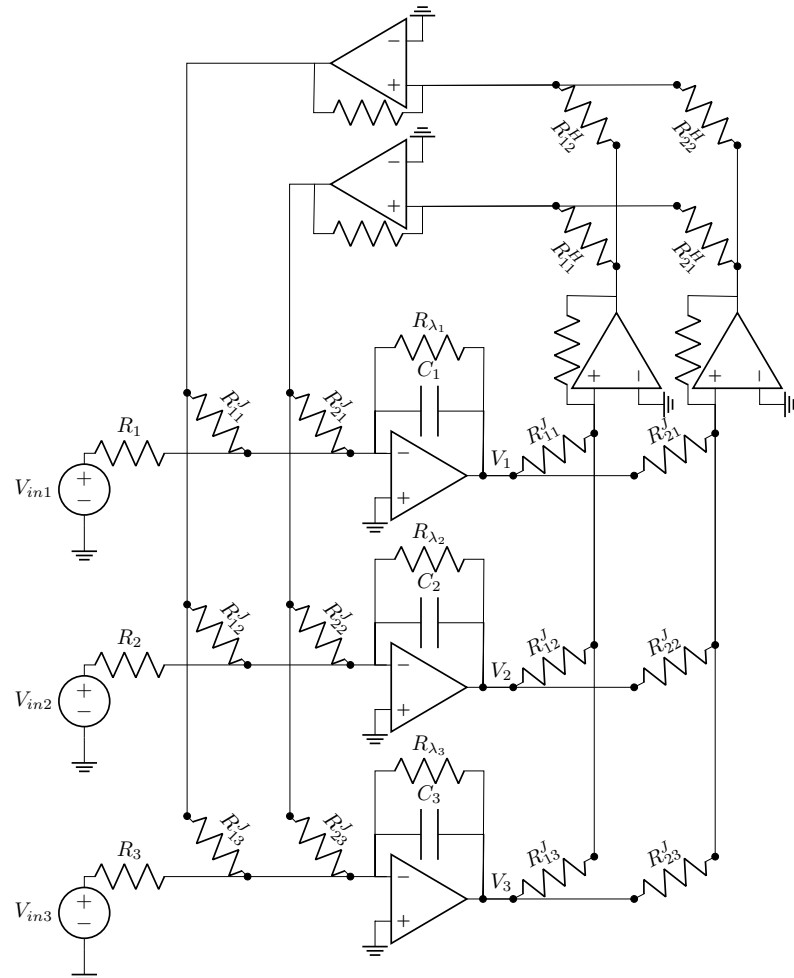

Let us consider the circuit diagram shown above, where $N = 3$, $b = 1$, $d_z = 2$. We assume the capacitors all have the same value $C$, and the resistors with no labels all have the same value $R_0$. By Kirchhoff's current law, we obtain the equation of motion for the voltage vector $V = (V_1, V_2, V_3)$ as:

$$C\dot{V} = -(\mathcal{G}V + \lambda V - R^{-1}V_{in})$$

with $V_{in} = (V_{in1}, V_{in2}, V_{in3})$, $R = \text{diag}(R_1, R_2, R_3)$, $\lambda = \text{diag}(1/R_{\lambda_1}, 1/R_{\lambda_2}, 1/R_{\lambda_3})$. In the case of TNGD we have

$$\mathcal{G} = J_f^T H_L J_f = \begin{pmatrix} \frac{1}{R_{11}^J} & \frac{1}{R_{12}^J} \\ \frac{1}{R_{21}^J} & \frac{1}{R_{22}^J} \\ \frac{1}{R_{31}^J} & \frac{1}{R_{32}^J} \end{pmatrix} \begin{pmatrix} \frac{1}{R_{11}^H} & \frac{1}{R_{12}^H} \\ \frac{1}{R_{21}^H} & \frac{1}{R_{22}^H} \end{pmatrix} \begin{pmatrix} \frac{1}{R_{11}} & \frac{1}{R_{21}} & \frac{1}{R_{31}} \\ \frac{1}{R_{12}} & \frac{1}{R_{22}} & \frac{1}{R_{32}} \end{pmatrix} \frac{1}{R_0^2}, \qquad (23)$$

where we therefore have one set of resistors $R^J$ representing the $J$ matrix and its transpose, and one set of resistors representing $R^H$ the H matrix. At steady state the average voltage vector corresponds to the natural gradient estimate, since for $\dot{V} = 0$, the average voltage vector is $\langle V \rangle = \mathcal{G}^{-1} R^{-1} V_{in}$, which corresponds to the solution of the linear system $Ax = b$ with $A = \mathcal{G}$, $x = V$, $b = R^{-1} V_{in}$. The resistor values $R_{ij}^J$ ($R_{ij}^H$) can directly be calculated as $1/J_{ij}$ ($1/H_{ij}$) (or $1/J_{ji}$ for the transpose), and the total number of resistors in the circuit is $(bd_z)^2 + 2bd_z N$ (16 in the schematic shown). This means that one may store $J_f$ and $H_L$ in memory, and send $J_f$ twice to the hardware (one to the left resistor array, once to the right resistor array).

One may run the thermodynamic linear solver by setting the voltage values $V_{\text{in}}$ to the the gradient $\nabla \ell$ with a digital-to-analog converter, and set the values of the programmable resistors thanks to a digital controller.

$N$ and $d_z$ enter the digital transfer time (since $2Nbd_z + (bd_z)^2$ numbers with a given precision have to be set on the device) and ADC time ($N$ numbers to send back the natural gradient estimate to the digital device), which are two contributions to the estimated runtime per iteration. The relaxation time of the system is

$$\tau = \frac{RC}{\alpha_{\text{min}}}$$

where $R$ is a resistance scale (which means that all resistances $R_{ij}$ are a multiple of this), $C$ is the capacitance (assuming all the capacitances are the same), and $\alpha_{\text{min}}$ is the smallest eigenvalue of the (unitless) $\mathcal{G}$ matrix. After this time, all the modes of the system will have relaxed, which may be too conservative (for example, in the case where there is only one slow mode, and all other modes are fast). With regularization, $\alpha_{\text{min}}$ is lower-bounded by the regularization factor $\lambda$ (which is $10^{-2}$ for the MNIST experiments, and $1$ for the language model fine-tuning experiments). For timing purposes, we kept $RC$ as the relaxation time, because of the problem-dependence of $\alpha_{\text{min}}$. To obtain the comparisons to other digital methods, we considered the following procedure to run the thermodynamic linear system on electrical hardware:

1. Digital-to-analog (DAC) conversion of the the gradient vector with a given bit-precision.

2. Set the configuration of the programmable resistors ($bd_z(bd_z + 2N)$) values with a given bit precision to set).

3. Let the dynamics run for $t$ (the analog dynamic time). Note that for experiments $t$ was chosen heuristically by exploring convergence in the solutions of the problem of interest.

4. Analog-to-digital (ADC) conversion of the solution measured at nodes $V_i$ to the digital device.

The runtime estimated are based on the following assumptions:

- 16 bits of precision.

- A digital transfer speed of 50 Gb/s.

- $R = 10^3 \, \Omega$, $C = 1 \, \text{nF}$, which means $RC = 1\mu s$ is the characteristic timescale of the system.

Finally, note that in all cases that were investigated, the dominant contribution to the total runtime of TNGD was the digital steps to compute the gradients, Jacobian and Hessian matrices. Hence some assumptions about the DAC/ADC may be relaxed and the total TNGD runtime would be similar. The $RC$ time constant may also be reduced to make the algorithm faster, although this is found to be easily sub-dominant with respect to input operations (setting the configuration of the device).

## D    SIMULATING TNGD

The results reported in this paper require simulating the thermodynamic device. To do so, we employ a Euler-Maruyama discretization of Eq. 15, where the update equation is:

$$\tilde{g}^{(k+1)} = \tilde{g}^{(k)} + \delta t(G\tilde{g}^{(k)} - \nabla \ell) + z\sqrt{2\kappa_0 \delta t} \tag{24}$$

where $\delta t$ is a step size, $z \sim \mathcal{N}(0, 1)$ and the GGN-vector product $G\tilde{g}^{(k)}$ is evaluated in linear time, with no need to construct $G$ as explained in Section 3. One may consider higher-order schemes, which in general will cost $d$ GGN-vector products (hence $2d + 1$ model calls, accounting for the gradient) for each step of an order $d$ solver.

With an Euler-Maruyama scheme, one therefore requires 3 model calls per time step, which results in long simulation times for the larger $t$ values we report.

# E  EXPERIMENTAL DETAILS

All experiments except the one reported in Fig. 2 were carried out on a Nvidia A100 GPU with 80 GB of RAM. The experiment corresponding to Fig. 2 was carried out on a AMD EPYC 7763 64-Core CPU with 32 GB of RAM (the results on the GPU had too much variance even for a large number of repetitions). For Fig. 2, $b = 32$, $c = 200$, and the results were obtained by repeating over 5 manual random seeds, with the standard deviation over runs being shown as shaded areas. Modifying $c$ has the simple effect of shifting the curve on the scale.

All experiments are written in PyTorch (Paszke et al., 2019), and we have used the `posteriors` library (Duffield et al., 2024) which supports GGN-vector products, constructing GGN matrices (for exact NGD and NGD-Woodbury) and the conjugate gradient solver in PyTorch.

## E.1  MNIST

For the MNIST experiments, we train on 10,000 images and test on 10,000 other images, with $b = 64$ for 10 epochs. Below is a table with hyperparameters for each figure in the main text:

| Figure – Optimizer | Optimizer parameters |
|---|---|
| Fig. 3 – Adam | $\eta = 0.001, \beta_1 = 0.9, \beta = 0.999, \epsilon = 1e-8$ |
| Fig. 3 – TNGD | $\eta = 0.01, \beta = 0, t = 50\tau, t_d = 0, \lambda = 0.01, \delta t = 0.1\tau$ |
| Fig. 2(a) – TNGD | $\eta = 0.01, \beta = 0, \lambda = 0.01, \delta t = 0.1\tau$ |

For these experiments, the plotted data is the mean (and standard deviation) of the moving average over 200 points for the five same manual random seeds. The Adam experiments took $\sim 5$ minutes to run, while the longest TNGD experiments took $\sim 14$ hours (due to many time steps being required, see Section D. We performed sweeps over the damping and learning rate value of TNGD which we do not report in the paper that took $\sim 10$ days of accumulated total runtime.

### E.1.1  NOISY SIMULATION

One key feature of TNGD is that it is noise-resilient. Indeed, because the solution of the linear system of equation 6 is encoded in the first moment of the equilibrium distribution, any noise that is approximately Gaussian will not affect much the quality of the results for reasonable noise levels. In Fig. 6, the loss vs. iterations are shown for varying noise levels, which are defined by the value of the noise variance $\kappa_0$. For $\kappa_0 < 0.01$, the noise essentially does not affect the performance of TNGD (as the influence of noise of performance starts to saturate at this value), even for a very small analog dynamics time (here, $t = \tau$). For a realistic electrical device where $\theta_t$ are voltages, the contribution of thermal noise to the noise level would be $\kappa_0 \sim 10^{-6}V$. Other noise sources may contribute to the noise level, but because of the nature of the TNGD algorithm, it exhibits a high noise-resilience. Note that it is also possible to collect more samples from the device to reduce influence of the noise if the noise level is large.

## E.2  EXTRACTIVE QUESTION-ANSWERING

For the QA experiments, we train on 800 articles and test on 200 other articles of the SQuaD dataset, with $b = 32$ for 5 epochs. Below is a table with hyperparameters for Fig. 5:

| Figure – Optimizer | Optimizer parameters |
|---|---|
| Fig. 5(a) – TNGD | $\eta = 0.01, \beta = 0, t = 0.4\tau, t_d = 0, \lambda = 1$ |
| Fig. 5(a) – Adam | $\eta = 0.001, \beta_1 = 0.9, \beta_2 = 0.999, \epsilon = 1e-8$ |
| Fig. 5(b) – TNGD-Adam | $\eta = 0.001, \beta_1 = 0, \beta_2 = 0, \epsilon = 1e-8, t_d = 0, \lambda = 1, \delta t = 0.02\tau$ |

We apply low-rank adaptation (LoRA) to the $Q, K, V$ modules and output projection matrices of the attention layers with parameters $r = 2, \alpha = 32$ and a dropout of 0.1. LoRA consists in replacing the pre-trained weight matrices of the targeted layers $W_0$ by:

$$\tilde{W} = W_0 + AB, \tag{25}$$

where $A$ and $B$ are two rectangular matrices with their smaller dimension being $\alpha$ (hence $AB$ is low-rank). We used the `peft` package (Mangrulkar et al., 2022), which interfaces smoothly with PyTorch and `posteriors`.

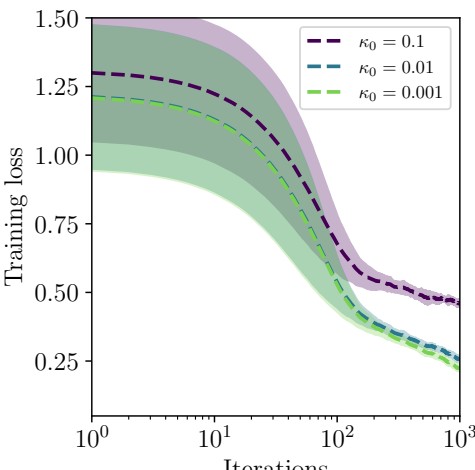

Figure 6: **Training loss vs. iterations for varying noise levels.** The noise level is defined by the noise variance $\kappa_0$ entering equation 15. Here $t = \tau$.

For these experiments we only report a single fixed random seed due to long simulation times. The Adam experiments took $\sim 20$ minutes, while the longest TNGD experiments took $\sim 2$ days (due to many time steps being required, see Section D. We performed sweeps over the damping and learning rate value of TNGD and Adam which we do not report in the paper that took $\sim 10$ days of accumulated total runtime.

