# OpenReview forum: "Thermodynamic Natural Gradient Descent"
_ICLR.cc/2025/Conference — ICLR 2025 Conference Withdrawn Submission_

### Official Review · Reviewer_uc62 · 2024-11-01

**Soundness:** 4
**Presentation:** 3
**Contribution:** 4
**Rating:** 8
**Confidence:** 3

**Summary:**

This paper introduces a hybrid digital-analog technique for computing natural gradients, leveraging the analog properties of a thermodynamic, stochastic system to quickly invert the Fisher matrix. This inverted matrix can then be used alongside the gradient of the loss function to descend a loss function more efficiently than by using the gradient alone.

**Strengths:**

I think this paper is very timely, and tackles an important topic. Using hardware (in this case, fundamental physics) to effectively compute will be essential to deep learning going forward. The paper is for the most part nicely written, and straightforward to follow. The experiments are convincing.

**Weaknesses:**

There are some terminological concerns/questions (below), which would improve the paper if addressed.

**Questions:**

- General terminology question: In the original Amari (1998) paper (page 4), the term 'natural gradient' refers to the expression $G^{-1} \nabla \mathcal{L}(w)$, where $G$ is an arbitrary symmetric positive definite matrix. In your paper (L127), you use 'natural gradient' exclusively for the case where $G$ is the Fisher information matrix. Which interpretation is correct?

- L077: Condition number of what? Additionally, NGD-CG is not defined until later in the paper, so avoid using this acronym here.
- L159: "(By curvature matrix, we mean any matrix capturing information about the loss landscape)." Could you clarify? Information about which aspect of the loss landscape?
- L196: I looked at the cited references for the claim about $\kappa$, but found no mention of the condition number of $F$. Also, since $F$ is not fixed, what do you mean by the condition number of F? An upper bound on the condition number across all possible $F$?
- The authors may be interested in [recent work](https://arxiv.org/abs/2409.16422), suggesting that any effective learning rule can be formulated as natural gradient descent. This might help provide additional motivation for efficient natural gradient computation?

---

### Official Review · Reviewer_hiMt · 2024-11-01

**Soundness:** 3
**Presentation:** 2
**Contribution:** 2
**Rating:** 3
**Confidence:** 3

**Summary:**

This paper proposes a method of reducing the per-iteration computational cost of NGD (a second order optimization method that has been shown to converge in less iterations than GD), using a hybrid analog-digital algorithm. The algorithm has a runtime that scales linearly with the number of parameters and can be made close to the runtime of first-order optimizers.

**Strengths:**

This paper proposes a new algorithm that combines analog and digital computation to reduce the per-iteration cost of NGD, and performed thorough comparison of the scaling of the runtime between their method and various existing methods. The motivation and the method are clearly conveyed. The authors also performed several numerical experiments comparing the existing methods and TNGD.

**Weaknesses:**

The paper could be written more clearly and concisely. I find the introduction of certain aspects too detailed and not necessarily helpful for understanding the authors’ contribution (such as the Fisher information etc.) and certain aspects too unclear for people not coming from an optimization background (such as the introduction of fast matrix vector product, the convergence speed and performance is only vaguely discussed). Also, this paper uses many different notations, and the introduction of these notations are often hidden in the text, making it very hard to read. I suggest that the author clearly and explicitly write out definitions and their claims.
While the paper proposes a new algorithm, it seems to be a direct combination of existing works on natural gradient descent and using an SPU to solve linear systems, although because I am not familiar with analog computing, I could be wrong about my judgement on the novelty and contribution of this work.
The paper would also benefit from more numerical comparisons, both on more datasets and also tuning different hyperparameters (such as the different hyperparameters in adam) to demonstrate the robustness of their findings.

**Questions:**

1.	It seems unnecessary to take the detour by introducing the statistics point of view and Fisher Information and then the GGN, as GGN is a relatively common approach and can be derived from the loss function without necessarily introducing the KL divergence. The rest of the paper also does not ever use the Fisher Information matrix again and only uses the GGN. I suggest that the authors introduces GGN right after the loss function and cite relevant references relating the GGN to the FI matrix.
2.	This is a minor point. The ‘GGN vector product Gv ‘ has not been previously introduced before line 191. I suppose it refers to product of GGN with an arbitrary vector v? Please state explicitly.
3.	What do you mean by “convergence in \sqrt{k} steps … is not required …”, do you mean that the algorithm does not necessarily converge in \sqrt{k} steps but achieve good performance despite not converging? This sentence is confusing to me.
4.	The authors switch between using G and F, to my understanding, the authors stick to the GGN approximation of the fisher information matrix, so I suggest that the authors stick to one notation.
5.	I am not familiar with analog computing. From my understanding it seems that for solving the linear system, both the conjugate gradient method and the authors’ thermodynamic natural gradient descent method can be understood as minimizing the same quadratic objective, only CG also takes orthogonal gradient steps at each iteration while the thermodynamic NGD is basically gradient descent plus thermal noise. Naively I would expect CG to converge in less steps than NGD, while the authors seem to suggest otherwise. I suppose the longer convergence time of NGD is manifested in this extra factor “t”? The authors then claim that t does not need to be very long for the performance to be good, however, this seems an empirical observation, it would be nice if the authors could provide some bounds on t to show that thermodynamic NGD is indeed superior.
6.	In Fig.3, test loss almost always seems to be smaller than training loss except maybe at the end, is this because of some regularization method or do you have any explanations for this?

---

### Official Review · Reviewer_6Z95 · 2024-11-04

**Soundness:** 2
**Presentation:** 2
**Contribution:** 1
**Rating:** 3
**Confidence:** 3

**Summary:**

The paper presents thermodynamics natural gradient descent, an approximation to natural gradient descent that, instead of inverting the matrix needed for NG updates, approximates the inverse via a random walk. This is claimed to be implementable in hardware, thus offloading the matrix inverse from GPUs.

**Strengths:**

1. The presented approach seems to be compatible with alternative (to GPUs) hardware, although I’m not familiar with thermodynamic computing to judge it properly.
2. The proposed natural gradient approximation is theoretically justified.

**Weaknesses:**

## Evaluation issues
While the paper’s approach is interesting, the evaluations could be significantly improved.

First, MNIST performance. Fig.3b shows that
1. Adam and TNGD achieve the same test error; (Line 367 claims “TNGD outperforms Adam”, and then line 371 claims TNGD achieves better test accuracy, while it is clearly not true in Fig3b.)
2. TNGD overfits significantly more than Adamw (0 train error vs. somewhat close to test error for Adam); (Line 368 claims TNGD “generalizes better”.)
3. Most importantly, both methods achieve about 94% test accuracy on MNIST. MNIST is an extremely easy dataset – pretty much any neural network should be able to get 97-99% test accuracy (https://paperswithcode.com/sota/image-classification-on-mnist?metric=Accuracy). As Sec. 5.1 uses a conv layer with two fc layers, it should be enough for good performance as well; this suggests the evaluation was not done correctly.

Second, NLP experiments in Sec. 5.2 only report the training loss (also, the TNGD-Adam combination presented there was not tested for MNIST).

Finally, TNGD is an approximation NGD, but no NGD baselines are shown.

With only two datasets and not convincing performance, it is hard to judge if the proposed approach (or using NGD at all) is worth the computational overhead.


## Justification for using NGD

I think the paper does not successfully justify using NGD for training neural networks. Its performance benefits are unclear from the presented results (although probably explored in other papers), but the computational overheads are very significant:
- Tab. 1 shows that TNGD is much more expensive that SGD;
- the suggested (future) approach would involve using dedicated hardware.

**Questions:**

1. Natural gradient descent can be linked to mirror descent (see e.g. https://franknielsen.github.io/blog/NaturalGradientConnections/NaturalGradientConnections.pdf). They’re not strictly identical, but mirror descent is a first order method with the same time complexity as gradient descent. If the goal is to improve performance by using better conditioned gradients, wouldn’t mirror descent be a (computationally) better target algorithm than NGD?
2. Eq. 12: the first $\mathbb{I}$ should be $v$.

---

### Note · Authors · 2024-11-26

**Comment:**

We thank the reviewers for the feedback and will revise our paper for a further submission.

**Withdrawal Confirmation:**

I have read and agree with the venue's withdrawal policy on behalf of myself and my co-authors.